# Structure, Physicochemical Properties and Biological Activity of Lipopolysaccharide from the Rhizospheric Bacterium *Ochrobactrum quorumnocens* T1Kr02, Containing d-Fucose Residues

**DOI:** 10.3390/ijms25041970

**Published:** 2024-02-06

**Authors:** Aleksandra A. Krivoruchko, Evelina L. Zdorovenko, Maria F. Ivanova, Ekaterina E. Kostina, Yulia P. Fedonenko, Alexander S. Shashkov, Andrey S. Dmitrenok, Elizaveta A. Ul’chenko, Oksana V. Tkachenko, Anastasia S. Astankova, Gennady L. Burygin

**Affiliations:** 1Department of Organic and Bioorganic Chemistry, Institute of Chemistry, Saratov State University, 410012 Saratov, Russia; 2N.D. Zelinsky Institute of Organic Chemistry, Russian Academy of Sciences, 119991 Moscow, Russia; zdorovenkoe@mail.ru (E.L.Z.);; 3Department of Plant Breeding, Selection, and Genetics, Faculty of Agronomy, Saratov State University of Genetics, Biotechnology and Engineering Named after N.I. Vavilov, 410012 Saratov, Russiaoktkachenko@yandex.ru (O.V.T.); 4Institute of Biochemistry and Physiology of Plants and Microorganisms, Saratov Scientific Centre of the Russian Academy of Sciences, 410049 Saratov, Russia; 5Department of Biochemistry and Biophysics, Faculty of Biology, Saratov State University, 410012 Saratov, Russia; 6Department of Biomedical Products, Faculty of Chemical Pharmaceutical Technologies, D.I. Mendeleev University of Chemical Technology of Russia, 125047 Moscow, Russia

**Keywords:** *Ochrobactrum quorumnocens* T1Kr02, lipopolysaccharide, O-polysaccharide structure, d-fucose, β-glucane, succinate, dynamic light scattering, supramolecular particles, potato, microplants, in vitro culture

## Abstract

Lipopolysaccharides (LPSs) are major components of the outer membranes of Gram-negative bacteria. In this work, the structure of the O-polysaccharide of *Ochrobactrum quorumnocens* T1Kr02 was identified by nuclear magnetic resonance (NMR), and the physical–chemical properties and biological activity of LPS were also investigated. The NMR analysis showed that the O-polysaccharide has the following structure: →2)-β-d-Fuc*f*-(1→3)-β-d-Fuc*p*-(1→. The structure of the periplasmic glucan coextracted with LPS was established by NMR spectroscopy and chemical methods: →2)-β-d-Glc*p*-(1→. Non-stoichiometric modifications were identified in both polysaccharides: 50% of d-fucofuranose residues at position 3 were O-acetylated, and 15% of d-Glc*p* residues at position 6 were linked with succinate. This is the first report of a polysaccharide containing both d-fucopyranose and d-fucofuranose residues. The fatty acid analysis of the LPS showed the prevalence of 3-hydroxytetradecanoic, hexadecenoic, octadecenoic, lactobacillic, and 27-hydroxyoctacosanoic acids. The dynamic light scattering demonstrated that LPS (in an aqueous solution) formed supramolecular particles with a size of 72.2 nm and a zeta-potential of –21.5 mV. The LPS solution (10 mkg/mL) promoted the growth of potato microplants under in vitro conditions. Thus, LPS of *O. quorumnocens* T1Kr02 can be recommended as a promoter for plants and as a source of biotechnological production of d-fucose.

## 1. Introduction

Polysaccharides with biological activity are produced by organisms of various taxa [1,2]: fungi (including chitin, β-glucans, and galactomannans [3,4,5]), plants (including pectins, carrageenans, and fucoidans [6,7,8,9,10]), and animals (including heparin and hyaluronic acid [11,12,13]). The most well-characterized bioactive polysaccharides are bacterial lipopolysaccharides (LPSs), for which different chemical structures, biochemical biosynthetic pathways, the genetic basis of variability, and mechanisms of biological activity have been described [14,15,16,17,18,19,20,21]. LPS molecules are amphiphilic because they contain hydrophobic (lipid A) and hydrophilic (O-polysaccharide) parts linked by an oligosaccharide core [16]. Such glycolipid molecules occupy most of the surface of the outer membrane of Gram-negative bacteria [22] and play an important role in the interaction of bacteria with environmental objects [23,24,25].

Despite the description of many structures of repeating units of O-polysaccharides in various Gram-negative bacteria, the LPSs of many taxonomic groups have not yet been studied. One of these poorly studied groups is bacteria of the genus *Ochrobactrum*, family *Brucellaceae*, for which the O-polysaccharide structures of only five strains have been described [26]. In 2020, based on a high percentage of average nucleotide identity, bacteria of the genus *Ochrobactrum* were combined with *Brucella* into the genus *Brucella*/*Ochrobactrum* [27]. However, these two groups of bacteria have significant differences between themselves, which allows some researchers to challenge their classification into a single genus. Recently, a number of species isolated from potato roots (*Ochrobactrum quorumnocens*) [28] and soil (*Ochrobactrum chromiisoli*) [29] have been described, with *Ochrobactrum* being the generic name used by the authors. The structure of LPS molecules can serve as the foundation for technologies for the swift identification of pathogenic *Brucella* and saprophytic *Ochrobactrum*, owing to the distinct characteristics of antigenic properties and monosaccharide and fatty acid composition, as well as the molecular genetic markers of LPS biosynthesis genes [30].

In addition to taxonomy, the study of LPS from bacteria of the genus *Ochrobactrum* has a number of other applied aspects. The LPS of these bacteria has been shown to influence the growth and development of plants [31,32]. It was also previously described that the O-polysaccharides of three *Ochrobactrum* strains contain residues of d-fucose (d-Fuc) [33,34] or its derivative (d-Fuc3NAcyl) [26]. d-Fuc is a relatively rare monomer in natural polysaccharides; thus, studying the glycans of *Ochrobactrum* spp. is very promising for describing the structure and properties of new bacterial polysaccharides.

In this work, we studied the LPS of the *Ochrobactrum quorumnocens* strain T1Kr02, isolated from potato roots, determining the structure of the oligosaccharide repeating unit of the O-polysaccharide, the composition of fatty acids in lipid A, as well as physicochemical properties and biological activity in relation to potato plants.

## 2. Results

### 2.1. Preparation and Composition of LPS

From 10 L of the bacterial culture of *Ochrobactrum quorumnocens* T1Kr02, 5.6 g of dry cell biomass was obtained, from which the LPS was isolated using water–phenolic extraction with a yield of 2.75%. The LPS of *O. quorumnocens* T1Kr02 contained 31.2% carbohydrates (relative to glucose), 0.66% phosphorus, 0.92% 2-keto-3-deoxyoctonate (KDO), and trace amounts of proteins and nucleic acids (<1%). The fatty acid analysis of the LPS showed the prevalence of 3-hydroxytetradecanoic, hexadecenoic, octadecenoic, lactobacillic, and 27-hydroxyoctacosanoic acids.

### 2.2. Structure of Repeating Units of O-Polysaccharide

The LPS was degraded under mild acidic conditions to give two polysaccharide fractions, which were isolated by GPC on Sephadex G-50. As a result, two fractions (PS-1 and PS-2, with yields of 3.0% and 10.5%, respectively) were isolated and studied. Sugar analysis of the PS-1 by GLC of the alditol acetates revealed fucose while, in the PS-2, the main sugar found was glucose. The d configuration of Fuc and Glc was determined by GLC using published methods following their release from the OPS by full acid hydrolysis.

The ^1^H and ^13^C NMR spectra of the *O. quorumnocens* T1Kr02 PS-1 showed signals of different intensities, thus indicating that the polysaccharide is structurally heterogeneous, inter alia due to non-stoichiometric O-acetylation (there were signals for the O-acetyl group at δ_H_ 2.16 and δ_C_ 21.7). O-deacetylation of the PS-1 resulted in elimination of heterogeneity. Particularly, the ^1^H NMR spectrum of the de-O-acetylated PS-1 (DPS, Figure 1, upper projections and Table 1) showed signals for a two-anomeric-proton singlet at δ_H_ 5.43 and a doublet at δ_H_ 4.60 (*J*_1,2_ 8 Hz) and a CH_3_-C group six-proton doublet at δ_H_ 1.26 (*J*_5.6_ 6 Hz) (H-6 of Fuc). The other signals were located in the region δ_H_ 3.6–4.4. The ^13^C NMR spectrum of the DPS (Figure 1, left projection and Table 1) contained signals for two anomeric carbons at δ 103.3 (Fuc*p*) and 109.0 (Fuc*f*), two CH3-C groups at δ 16.7 (C-6 of Fuc*p*) and 19.5 (C-6 of Fuc*f*), and other sugar carbons in the region δ 68.6–90.5.

The ^1^H NMR spectrum of the DPS (Table 1) was assigned using 2D ^1^H,^1^H COSY, ^1^H,^1^H TOCSY, and ROESY experiments. The spectra revealed β-fucofuranose (β-Fuc*f*, residue A in Table 1) and β-fucopyranose (β-Fuc*p*, residue B in Table 1). Conclusions about sugar composition, ring size, and anomeric configuration (Formula and Table 1) were made based on the comparison of visible coupling constants and chemical shifts of the sugar residues and corresponding parent pyranoses [35,36,37] and furanoses [35,36,38,39].

The ^13^C NMR signals were assigned using 2D heteronuclear ^1^H,^13^C spectra HSQC [38] and HMBC. The assignment revealed positions of substitution in the residues at ^13^C chemical shifts’ comparison of the residues and corresponding parent sugars and taking into account well-known glycosylation effects [40,41]. The residue A (β-Fuc*f*) was found substituted at position 2 and residue B (β-Fuc*p*) proved to be substituted at position 3 (formula and Table 1).

The spectrum ROESY (Table 2) contained inter-residue peaks H-2A, C-1B; H-3A, C-1B; and H-3B, C-1A demonstrating a linear polysaccharide chain. The HMBC spectrum (Figure 2) finally confirmed the chain as the following:

Based on these data, it was concluded that the disaccharide repeating unit of the DPS of *O. quorumnocens* T1Kr02 has the following structure:→2)-β-d-Fuc*f*-(1→3)-β-d-Fuc*p*-(1→  **A**      **B**

The ^1^H and ^13^C NMR spectra of the initial PS-1 were assigned and analyzed as described above for the DPS (Table 1 and Table 2). A comparison of the ^1^H,^13^C HSQC spectra of the initial PS-1 and DPS showed a downfield displacement of a part of the Fuc*f* (**A**) H-3/C-4 cross-peak from δ 4.10/77.9 in the DPS spectrum to δ 5.04/79.3 in the OPS spectrum, thus indicating O-acetylation of Fuc*f* (**A**) in position 3 in the OPS. This conclusion was confirmed by upfield displacements (β-effect of O-acetylation [42]) of the signals for H-2/C-2, H-4/C-4, H-5/C-5 of Fuc*f* (**A**) from 4.34/90.5, 3.88/88.4, 3.94/68.6, respectively, to 4.46/87.0, 4.14/87.3, 4.02/68.3, respectively. As judged by the relative intensities of the NMR signals for the O-acetylated and the corresponding non-O-acetylated monosaccharides, the degree of O-acetylation was ~50%. Thus, PS-1 is an O-specific polysaccharide of *O. quorumnocens* T1Kr02. To our knowledge, it is unique among the known bacterial polysaccharide structures.

Molecular modeling of the structure of the *O. quorumnocens* T1Kr02 PS-1 fragment (Figure 3) showed that the (1→3)-β-linkage between the d-Fuc*f* and d-Fuc*p* residues of the repeating units provides a reversal in the arrangement of monosaccharide residues within the O-polysaccharide. As a result of this rotation, the next two monosaccharide residues are located in a plane 90° deviated from the plane with the two previous residues. This organization of the molecule can lead to a denser arrangement of O-polysaccharide chains in space on the surface of cells. The acetate group is oriented outside the helix-like polysaccharide structure formed (Figure 3b and Appendix A). The external location of the acetate group can promote the formation of additional hydrophobic bonds, both between two O-polysaccharides and with other macromolecules.

The ^13^C NMR spectrum of the PS-2 (Figure 4, left projection, Table 1) contained two sets of signals. The major set was typical for a regular glucan and contained six signals including an anomeric one at δ_C_ 103.4. The minor set contained additionally four peaks of a succinic acid residue (Table 1).

The ^1^H NMR spectrum of the PS-2 (Figure 4, upper projection, Table 2) was also a heterogeneous one. The main set of signals contained intense peaks of the glucan and the minor set showed additional signals of the succinic acid.

The ^1^H NMR spectrum was assigned using 2D COSY and TOCSY spectra (Table 1).

The HSQC spectrum (Figure 4) revealed β-configuration of the glucopyranose in the glucan and substitution in the residues at position 2 (the main set of signals) and positions 2 and 6 (the minor set of signals), taking into account the downfield position of the signals C-2 and C-6 as compared with that of the β-glucopyranose.

The HMBC spectrum (Figure 5) confirmed the conclusion due to the presence of the inter-residue peaks Glc H-1/Glc C-2, Glc H-2/Glc C-1, and Glc H-6,6′/Suc C-1.

Taking NMR chemical shifts and sugar analysis data together, it was suggested that the PS-2 is a homopolymer of glucose substituted in position 6 by succinate (~15%) with the following structure:→2)-β-d-Glc*p*-(1→
Suc(⌋6_(~15%)_

A comparison of the HSQC spectra of PS-2 and *Sinorhizobium meliloti* 1021 [42] showed their identity. Based on this, we can suggest that isolated PS-2 is also a cyclic β-(1→2) glucan.

Molecular modeling of the structure of the *O. quorumnocens* T1Kr02 PS-2 fragment (Figure 6) showed that d-Glc*p* residues presumably form a dense helix-like structure. The succinate residues are oriented outward from the glucan (Figure 6b and Appendix A). It may contribute to the better dissolution of PS-2 in water and the formation of additional hydrogen and electrostatic bonds with other molecules.

### 2.3. Physicochemical Properties of LPS in the Aquatic Medium

LPS are amphiphilic molecules and, accordingly, supramolecular particles are formed in water. The results of measurements using the dynamic light-scattering method (Figure 7) showed that in an aqueous solution, *O. quorumnocens* T1Kr02 LPS particles had a diameter of 72.2 ± 3.6 nm and a zeta-potential of –21.5 ± 0.7 mV.

This is the first work that presents the characteristics of supramolecular particles of LPS from bacteria of the genus *Brucella*/*Ochrobactrum* in an aquatic environment. Both the size and zeta-potential of *O. quorumnocens* T1Kr02 LPS micelles can serve as a model for comparison when describing the physicochemical properties of LPS from other strains of this group of bacteria.

Electrophoretic separation in a polyacrylamide gel (Figure 8) demonstrated the predominance of the high-molecular-weight fraction of molecules in the LPS composition of *O. quorumnocens* T1Kr02 in the absence of heterogeneity, manifested in SDS-PAGE as a ladder band profile, characteristic of enterobacterial LPS.

### 2.4. Biological Activity of the LPS towards Potato Microplants

The addition of 10 μg/mL LPS of strain T1Kr02 to the cultivation medium had a positive effect on the growth of potato microplants of the cultivar Kondor (Table 3). The length of shoots increased by 35% (*p* ≤ 0.05) and the dry weight of shoots by 76% (*p* ≤ 0.05). LPS of strain T1Kr02 had an ambiguous effect on the development of the root system. The number and length of roots did not change significantly relative to the control variant, while the wet weight of the roots increased by 89% (*p* ≤ 0.01) and the dry weight by 57% (*p* ≤ 0.05). Thus, under the influence of LPS of strain T1Kr02, potato microplants formed roots with a greater mass per unit length.

## 3. Discussion

In this work, an LPS preparation was isolated from the dry biomass of *O. quorumnocens* T1Kr02 cells with a yield of 2.75%. For previously studied strains of *Ochrobactrum* spp., LPS yield was reported to be 8.7% for *O. cytisi* strain IPA 7.2 [32], 6.5% for *O. endophyticum* strain KCTC 42485 [26], and 2.35% for *O. rhizosphaerae* strain PR17 [34].

The fatty acid analysis of the LPS showed the prevalence of 3-hydroxytetradecanoic, hexadecenoic, octadecenoic, lactobacillic, and 27-hydroxyoctacosanoic acids. These data confirm the presence of lactobacillic and 27-hydroxyoctacosanoic acid residues in lipid A of *Ochrobactrum* spp. and binding with amide bonds, as was previously described for the *O. anthropi* strain LMG3331 [43]. A difference in the fatty acid composition of lipid A of *O. quorumnocens* T1Kr02 is the absence of 3-hydroxyhexadecanoic and 3-hydroxyoctadecanoic acid residues, the presence of which was described for strains *O. anthropi* LMG3331 and *O. intermedium* LMG3301 [43] and *O. cytisi* IPA 7.2 [32]. The species *Ochrobactrum quorumnocens* together with the species *O. rhizosphaerae*, *O. chromiisoli*, *O. pseudogrignonense*, *O. thiophenivorans*, *O. grignonense*, and *O. pituitosa* within the *Brucella/Ochrobactrum* group form Clade 2 [44] or group *Ochrobactrum*_A (GTDB-Tk, https://github.com/kbaseapps/kb_gtdbtk accessed 16 January 2024) (Appendix A), phylogenetically remote from other *Ochrobactrum* spp. (*O. anthropi*, *O. cytisi*, *O. intermedium*, and *O. oryzae*) and *Brucella* spp. (*B. canis*, *B. ovis*, and *B. suis*). This is the first work to describe the fatty acid composition of lipid A of bacteria from the *Ochrobactrum*_A group, which includes the species *O. quorumnocens*.

Mild acid hydrolysis of LPS yielded two polysaccharides (PS-1 and PS-2) in yields of 3.0% and 10.5%, respectively. For high-molecular-weight fractions of the *O. rhizosphaerae* strain PR17, the corresponding yields were 14% and 7% [34]. For phylogenetically more distant strains of *Ochrobactrum* spp., polysaccharide yields were significantly higher: 28% for *O. anthropi* LMG 3331 [45], 21% for *O. cytisi* IPA7.2 [32], 26.7% for *O. endophyticum* KCTC 42485 [26], 22% and 25% for two fractions of *O. cytisi* strain ESC1 [33].

We described PS-1 as an O-polysaccharide of the *O. quorumnocens* strain T1Kr02. PS-1 is unique among the known bacterial polysaccharide structures containing d-Fuc*p* and d-Fuc*f* residues, with 50% O-acetylation of d-Fuc*f*. d-Fuc is a monosaccharide that is relatively rare as a component of bacterial polysaccharides. d-Fuc*p* in the α-configuration has been identified in several bacterial O-polysaccharides, including bacteria of the genus *Ochrobactrum* [33,34]. In the β-configuration as part of O-specific polysaccharides, d-Fuc*p* was described for *Pantoea agglomerans* strain P1a [46]. In addition, the d-Fuc*p* derivative β-d-Fuc*p*3NAcyl was previously described by us for the O-polysaccharide of the *O. endophyticum* strain KCTC 42485 [26].

d-Fuc*f* residues, which are more rarely found in nature, were previously identified in the α-configuration in the O-polysaccharides of *Pseudomonas syringae* pv. *phaseolicola* GSPB 1552 [47] and *Eubactrium saburreum* T15 [48]. Meanwhile, β-d-Fuc*f* residues were described in the composition of O-polysaccharides in *E. coli* O52 [49] and *Pectinatus cerevisiiphilus* ATCC 29359 [50]. In *E. coli* O52, dTDP-D-Fuc*f* was shown to be synthesized from dTDP-d-Fuc*p* by the Fcf2 mutase [49] (Appendix A). Since the genome of strain T1Kr02 has not yet been deciphered, we do not know the genetics of the O-polysaccharide biosynthesis of this strain. However, based on the monosaccharide composition, we predict that the O-antigen biosynthesis gene cluster of the strain T1Kr02 contains genes for the biosynthesis of nucleotide-activated d-fucopyranose (*rmlA*, *rmlB*, and *fcf1*) and d-fucofuranose (*rmlA*, *rmlB*, *fcf1*, and *fcf2*). Thus, a “UDP-galactopyranose mutase” (acc. number KAA9371156) with 51% identity and 69% similarity to Fcf2 of the *E. coli* O52 (acc. number AAS99162) was identified in the genome of the *Ochrobactrum quorumnocens* strain RPTAtOch1 (acc. number genome VYXQ01000001) by the BLAST method. The *O. quorumnocens* strain RPTAtOch1 was isolated as an endophyte of the plant *Allium triquetrum* L. In the genome of this strain, CDS KAA9371156 is encoded close to the Fcf1 (acc. number KAA9371157) and glycosyltransferase (acc. number KAA9371155) genes (Appendix A), which suggests the possibility of the biosynthesis of d-Fuc*f* and the inclusion of its residues in the O-polysaccharide of strain RPTAtOch1, similar to the O-polysaccharide of T1Kr02.

This work reports for the first time that a bacterial polysaccharide contains both d-Fuc*f* and d-Fuc*p* residues. However, if dTDP-d-Fuc*p* is also a precursor of dTDP-d-Fuc*f* in strain T1Kr02, then the mechanism of incorporation of both forms of d-fucose, both pyranose and furanose, into the repeating unit of the O-polysaccharide remains unclear, since the glycosyltransferase that ensures the inclusion of the d-Fuc*p* residue into the oligosaccharide unit competes with Fcf2 for dTDP-d-Fuc*p* as a substrate (Appendix A). We assume that the enzyme activity of the Fcf2 in T1Kr02 cells is significantly lower than the activity of the Fcf1 enzyme, which could lead to the maintenance in the cell of a concentration of dTDP-d-Fuc*p* sufficient for the work of glycosyltransferase, which attaches the pyranose form of d-fucose to the repeating unit. At the same time, the glycosyltransferase activity for d-Fuc*p* must be close to the activity of Fcf2 in order for the biosynthesis of the furanose form (dTDP-d-Fuc*f*) to occur. In *E. coli* O52, the kinetic constants of the enzymes were *K_M_* = 0.38 mM, *V_max_* = 0.015 mmol/min for Fcf1 and *K_M_* = 3.43 mM, *V_max_* = 0.012 mmol/min for Fcf2 [49]. Clarification of these values for strain T1Kr02 and their correlation with the activity of glycosyltransferases requires a special biochemical study.

In addition, acetylated (4O*Ac*) and methylated (2O*Me*) d-Fuc residues were earlier described in the Nod factors of *Mesorhizobium loti* and *Bradyrhizobium japonicum*, respectively. These d-Fuc modifications are important for normal bacterial nodulation of lotus [51] and Afghan pea [52] roots. Our work describes for the first time the O acetylation of β-d-Fuc*f* residues at position C3.

PS-2 from *O. quorumnocens* T1Kr02 is periplasmic β-(1→2)-d-glucan, in which 15% residues were substituted in position 6 by succinate. It is well known that glucans are non-specific polysaccharides produced by different groups of organisms: plants, weeds, fungi, and Gram-positive and Gram-negative bacteria [42]. Thus, β-(1→2)-d-glucan was described together with the O-polysaccharide of strain *O. rhizosphaerae* PR17 [34]. The periplasmic glucans are cyclic and have a low molecular mass (6–40 glucose residues) [42]. The periplasmic glucans play an important role in the osmotic pressure of the cell. Cell-bound β-glucans may influence the virulence of Gram-negative bacteria and play an important role in osmotic adaptation in bacteria [42]. The presence of the β-glucan in the LPS of *O. quorumnocens* T1Kr02 could be explained by the coextraction of periplasmic glucans from the cells by hot water–phenol.

The carbohydrate content, measured using the phenol–sulfuric acid reaction, in the LPS preparation of *O. quorumnocens* T1Kr02 was 31% (relative to glucose). The weak extinction of the reaction products can be explained by the relatively low content of high-molecular-weight fractions of polysaccharides in the preparation and the weak reactivity of fucose, the products of which have extinction at 490 nm of about 40% relative to glucose [53].

In aqueous solution, LPS of *O. quorumnocens* T1Kr02 formed supramolecular particles with a diameter of 72.2 ± 3.6 nm and a zeta-potential of −21.5 ± 0.7 mV. Both characteristics of the particles are similar to those of the LPS of *O. cytisi* strain IPA7.2 (*d* = 66 nm; zeta-potential = −22 mV; unpublished data). Stronger negative values of the zeta-potential of supramolecular particles of LPS of strains *O. cytisi* IPA7.2 and *O. quorumnocens* T1Kr02 relative to LPS of the strain *O. anthropi* LMG3331 (−11.9 mV) [43] are explained by the presence of negatively charged groups: d-Man*N*AcA residues for *O. cytisi* IPA7.2 [32] and succinate residues for *O. quorumnocens* T1Kr02. In this case, the diameter of LPS micelles from *O. quorumnocens* T1Kr02 was significantly larger than the diameter of LPS micelles from bacteria of the genus *Azospirillum* with similar zeta-potential values [54].

The results of SDS-PAGE revealed the high-molecular-weight fraction of LPS, which may indicate the functioning of the methyltransferase FkbM, which controls the length of synthesized O-polysaccharides, similar to what we previously described for LPS of the *O. cytisi* strain IPA7.2 [32]. The methyltransferase gene FkbM is present in the O-antigen biosynthesis gene cluster of *O. quorumnocens* strain RPTAtOch1 (Appendix A), phylogenetically close to strain T1Kr02. And we assume that a similar gene is located in the gene cluster for O-polysaccharide biosynthesis in the *O. quorumnocens* T1Kr02 genome.

In the literature, LPS in relation to plants is mainly considered a bacterial elicitor of phytoimmunity [55,56,57], leading to growth inhibition. However, there are publications about the positive effect of LPS on plant growth and development [58]. Thus, for common wheat, it was shown that LPS of plant-growth-promoting rhizobacteria *Azospirillum* spp. improved seedling growth [59], increased grain yield [60], and benefited the yield of morphogenic tissue culture calluses [54]. The molecular mechanisms of interaction between bacterial LPS and plant cells are still poorly studied. To date, only two plant receptors have been described to participate in the formation of responses to LPS or lipid A: in *Arabidopsis thaliana*, the bulb-type lectin receptor-like kinase AtLORE [61], and in *Oryza sativa*, the LysM-type receptor-like kinase OsCERK1 [62]. At the same time, it is assumed that plants also have other receptors for LPS [23]. One of the mechanisms of specific plant responses to LPS from phytopathogenic or growth-promoting bacteria may be differences in changes in peroxidase activity and the functioning of calcium channels [63].

In relation to potato, the effects of LPS have been studied mainly for the phytopathogenic bacteria *Pseudomonas solanacearum* [64] and *Ralstonia solanacearum* [65]. Earlier, we established the plant-growth-promoting activity of LPS of the *O. cytisi* strain IPA7.2 [32] towards potato microplants cv. Kondor. As in this work, LPS from the *O. cytisi* strain IPA7.2 promoted shoot growth to a greater extent than the number and length of roots. Thus, LPS from *O. quorumnocens* strain T1Kr02 improves the development of microplants in vitro and can be used to increase the efficiency of microclonal propagation of potato.

LPS and O-polysaccharide of strain T1Kr02, due to their unique chemical composition and structure, can potentially also find applications in industrial biotechnology. Polymers containing 6-deoxyhexose residues can be used due to their rheological properties such as gelation, viscosity, or bioemulsification, and they also represent a source of some important monosaccharides [66]. The industrially important properties of d-Fuc-containing polysaccharides are still poorly understood, but these glycans can become a profitable source of various forms of d-Fuc and modified derivatives.

## 4. Materials and Methods

### 4.1. Preparation and Characterization of the Qualitative Composition of Lipopolysaccharide and O-Polysaccharide

The culture of the strain *Ochrobactrum quorumnocens* T1Kr02 (IBPPM 604, RCAM 04486) was from the Collection of Rhizosphere Microorganisms, Institute of Biochemistry and Physiology of Plants and Microorganisms, Russian Academy of Sciences (Saratov, Russia) (https://collection.ibppm.ru/ accessed 16 January 2024). Bacteria were grown in liquid malate–salt medium [67] at a temperature of 30 °C with a rotation of 120 rpm for 18 h; the bacterial suspension had an optical density of approximately 1.2 at 660 nm. The cells were repeatedly washed with PBS and acetone to obtain dry biomass. The biomass (5.6 g) was extracted by the Westphal procedure [68]. Proteins and nucleic acids were precipitated from the extract with CCl_3_COOH (pH 2.7) [69] and removed by centrifugation. The LPS preparation was lyophilized and stored at room temperature.

The carbohydrate and KDO contents were assessed by the phenol–sulfuric acid reaction at 490 nm [54] relative to glucose extinction and microassay at 548 nm [70], respectively. The phosphorus content was determined by the molybdenum blue method [71] after 15 min of incubation at 100 °C with the reagent (an ascorbic acid solution and an acidic mixture of molybdenum). The optical density of solutions was measured on a Specord 250 spectrophotometer (Analytik, Jena, Germany) in a cuvette with an optical path length of 5 cm at a wavelength of 882 nm. A calibration curve was constructed for different concentrations of potassium phosphate.

The LPS (100 mg) was hydrolyzed with 2% AcOH at 100 °C for 3 h. A lipid precipitate was removed by centrifugation (13,000× *g*, 20 min), and the carbohydrate portion was fractionated by gel-permeation chromatography on a column (56 × 2.6 cm) of Sephadex G-50 Superfine in 0.05 M pyridinium acetate buffer (pH 4.5). Elution was monitored with a Knauer differential refractometer (Knauer, Berlin, Germany).

Fatty acid methyl esters were obtained as described [72] and were analyzed on a GC-2010 (Shimadzu, Kyoto, Japan) chromatograph equipped with an EQUITY-1 (30 m × 0.32 mm) (Sigma-Aldrich, St. Louis, MO, USA) column using the temperature program of 110 °C for 5 min to 290 °C at 5 °C/min, final time 30 min; helium was used as a carrier gas at a flow rate of 1.3 mL/min. The evaporator temperature was 260 °C and flow distribution was 1:50. Fatty acids were identified using the standard mixture of the fatty acid methyl esters (Sigma Aldrich, USA).

### 4.2. Sugar Analysis

An OPS sample (0.5 mg) was hydrolyzed with 2 M CF_3_COOH (120 **°**C, 2 h), and the monosaccharides were analyzed by GLC as the alditol acetates [73] on a Maestro 7820 GC instrument (Interlab, Moscow, Russia) equipped with an HP-5ms column using a temperature program of 160 (1 min) to 290 °C at 7 °C/min. The absolute configuration of Gal was determined by GLC of the acetylated glycosides with (*S*)-2-octanol [74].

### 4.3. NMR Spectroscopy

An OPS sample was deuterium-exchanged by freeze-drying twice from 99.9% D_2_O and then examined as a solution in 99.95% D_2_O or a 9:1 D_2_O/H_2_O mixture. The ^1^H NMR and ^13^C NMR spectra were recorded using a Bruker Avance II 600 MHz spectrometer (Germany) at 30 °C using internal sodium 3-trimethylsilylpropanoate-2,2,3,3-d_4_ (δ_H_ 0.0, δ_C_ −1.6) as reference for calibration. The 2D NMR experiments were performed using standard Bruker software (Bruker TopSpin 2.1 program) and TopSpin 3.6.0 (academic license) were used to acquire and process the NMR data. A spin-lock time of 60 and a mixing time of 150 ms were used in the 2D ^1^H,^1^H TOCSY and ^1^H,^1^H ROESY experiments, respectively. The 2D ^1^H,^13^C HMBC spectrum was recorded with a 60 ms delay for evolution of long-range couplings.

### 4.4. Modeling of Oligosaccharide Structures

Structure formulas and molecular models of oligosaccharides were constructed using the Glycan Builder resource (http://csdb.glycoscience.ru/snfgedit/snfgedit.html accessed 16 January 2024) [75] for repeating units identified using NMR spectroscopy with different degrees of polymerization.

### 4.5. SDS-PAGE Electrophoresis

One hundred micrograms of LPS was dissolved in 100 μL of sample buffer (composition (per 100 mL): glycerol—15 mL; Tris—0.76 g; SDS—2.3 g; bromophenol blue—1.0 mg; 2-mercaptoethanol—10 mL; pH 6.8), after which the solution was incubated at 100 °C for 10 min. Aliquots of 40 µL, 20 µL, and 10 µL of the resulting solution were added to the corresponding tracks. SDS-PAGE electrophoresis was performed on a 15% polyacrylamide gel [76] at a voltage of 200 V. Staining was carried out with silver nitrate after oxidation with sodium periodate for 5 min [77].

### 4.6. Dynamic Light Scatting

For an aqueous solution of LPS with a concentration of 1.0 mg/mL, the size of supramolecular particles and their zeta-potential were measured by dynamic light scattering using a Malvern Nano-ZS zeta sizer (Malvern, UK) at a temperature of 37 °C as described earlier [54]. Ten measurements of each parameter were carried out in triplicate. The mean and confidence interval were calculated at *p* = 0.05 (95% confidence level).

### 4.7. Determination of the Biological Activity of LPS towards Potato Microplants

The effect of LPS from *Ochrobactrum quorumnocens* strain T1Kr02 on plants was studied as previously described [30]. Potato microplants (*Solanum tuberosum* L.) cultivar Kondor from the in vitro plant collection of the Department of Plant Breeding, Selection, and Genetics, Faculty of Agronomy, Vavilov University (Saratov, Russia), were used in this work. Sterile plants were fragmented into cuttings containing one node and one leaf. Each cutting was placed in a separate 25 mL tube (*d* = 15 mm) containing 6 mL of liquid Murashige–Skoog medium [78]. The tubes were incubated at 25 °C with 60 mM/(cm^2^ × s) illumination. Then, 60 μL of LPS solution (concentration 2 mg/mL) was added to the cultivation medium of 10-day-old microplants to obtain a final concentration of 10 μg/mL. Microplants on a nutrient medium without LPS were used as a control. Microplants were cultivated for another 20 days under the same conditions. In 30-day-old microplants, shoot length, number of nodes, number of roots, root length, and wet weight of shoots and roots were measured. To determine the dry weight of shoots and roots, they were dried in a thermostat at 105 °C to a constant weight. The control and experimental variants consisted of 30 microplants (*n* = 30). Average values were calculated from the results of three independent experiments. The significance of differences between variants was assessed using a *t*-test with confidence levels of 90%, 95%, and 99%.

## 5. Conclusions

Herein, we isolated and studied some properties of the LPS from the rhizosphere bacterium *Ochrobactrum quorumnocens* T1Kr02. It was established that the O-polysaccharide of this strain is a unique linear polysaccharide containing alternating d-fucopyranose and d-fucofuranose residues that are rarely found in natural polysaccharides. Thus, this work confirmed the prospect of studying O-polysaccharides of non-pathogenic strains of *Ochrobactrum* spp. to characterize novel d-Fuc-containing polysaccharides. The genetics and biochemistry of the biosynthesis of such glycans should be thoroughly studied in the future, as well as their potential as bioactive substances in humans, animals, and plants. It seems important to study the activities of enzymes involved in the biosynthesis of d-fucose residues to understand the consistency of bacterial production of d-fucopyranose and d-fucofuranose and their introduction into the repeating units of O-polysaccharide. Such data can become the basis for the effective biotechnological production of promising d-Fuc-containing glycans. Furthermore, the O-polysaccharide of *Ochrobactrum quorumnocens* strain T1Kr02 may be of interest as a source of raw material for the biotechnological production of d-Fuc. In this regard, the sequencing of the nucleotide sequence of the O-antigen biosynthesis gene cluster of strain T1Kr02, the study of the activity of enzymes for the biosynthesis of nucleotide-activated d-Fuc residues, and the optimization of biotechnological processes for the production of d-Fuc-containing polysaccharides are of interest in the future.

## Figures and Tables

**Figure 1 ijms-25-01970-f001:**
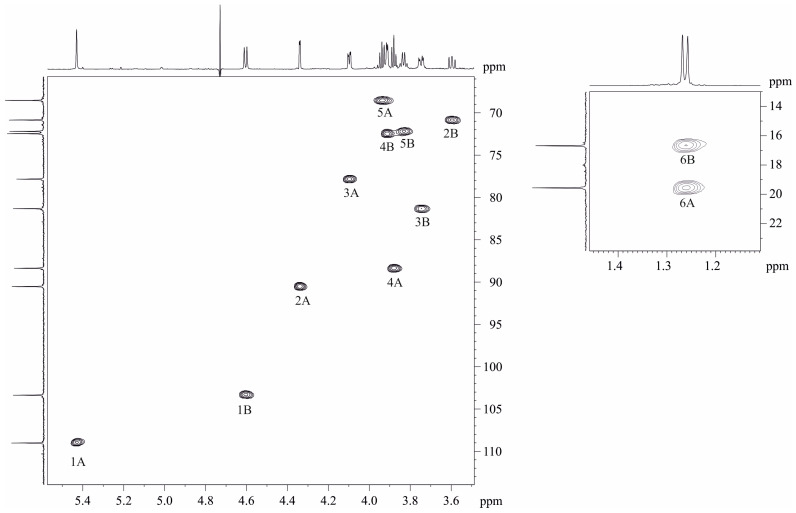
Part of a 2D ^1^H,^13^C HSQC spectrum of the de-O-acetylated PS-1 (DPS) of *O. quorumnocens* T1Kr02. The corresponding part of the ^1^H NMR spectrum is displayed along the horizontal F2 axis. Arabic numerals refer to the H/C pairs in the sugar residues denoted by letters as shown in Table 1.

**Figure 2 ijms-25-01970-f002:**
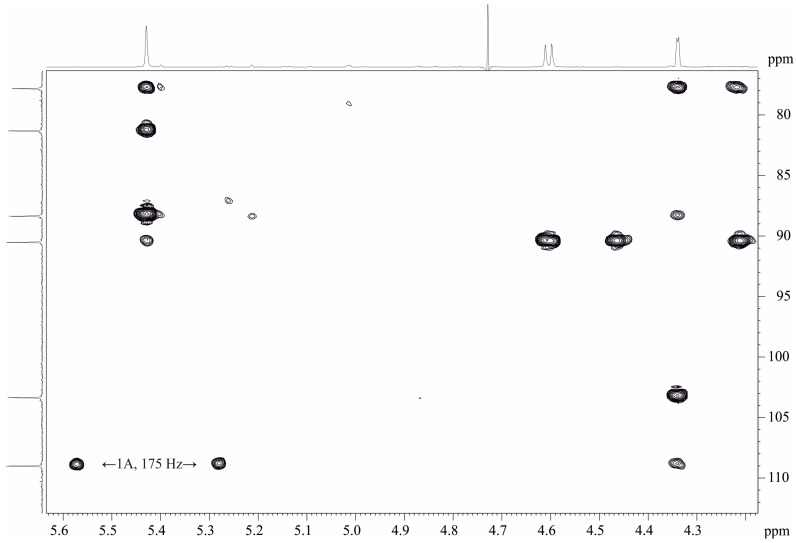
Part of a 2D ^1^H,^13^C HMBC spectrum of the de-O-acetylated PS-1 (DPS) of *O. quorumnocens* T1Kr02. The corresponding part of the ^1^H NMR spectrum is displayed along the horizontal F2 axis. Arabic numerals refer to the H/C pairs in the sugar residues denoted by letters as shown in Table 1.

**Figure 3 ijms-25-01970-f003:**
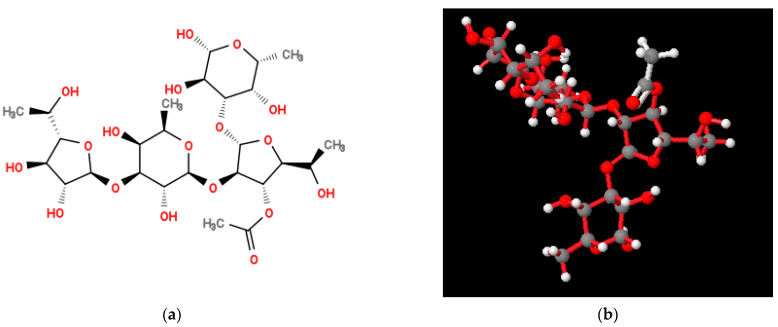
Structural formula (**a**) and molecular model (**b**) of two putative repeating unit of *O. quorumnocens* T1Kr02 O-polysaccharide (PS-1), containing d-Fuc*f* and d-Fuc*p* residues. One of the d-Fuc*f* residues (50%) at position 3 is O-acetylated. Atoms (balls) are designated as follows: gray—carbon; red—oxygen; white—hydrogen. Bonds between atoms in the oligosaccharide are indicated by red sticks, and bonds in the acetyl group are indicated by white sticks.

**Figure 4 ijms-25-01970-f004:**
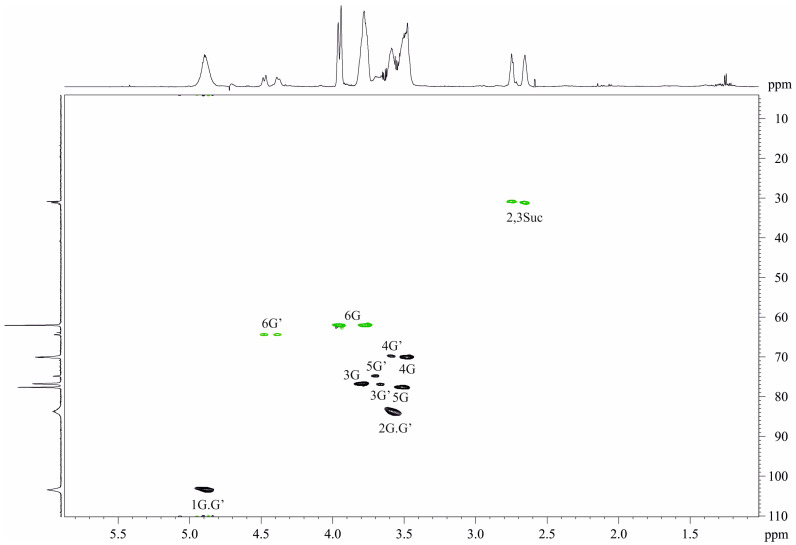
Part of a 2D ^1^H,^13^C HSQC spectrum of the PS-2 of *O. quorumnocens* T1Kr02. The corresponding part of the ^1^H NMR spectrum is displayed along the horizontal F2 axis. Arabic numerals refer to the H/C pairs in the sugar residues denoted by letters as shown in Table 1. Carbon atoms bearing even number of protons designed in green.

**Figure 5 ijms-25-01970-f005:**
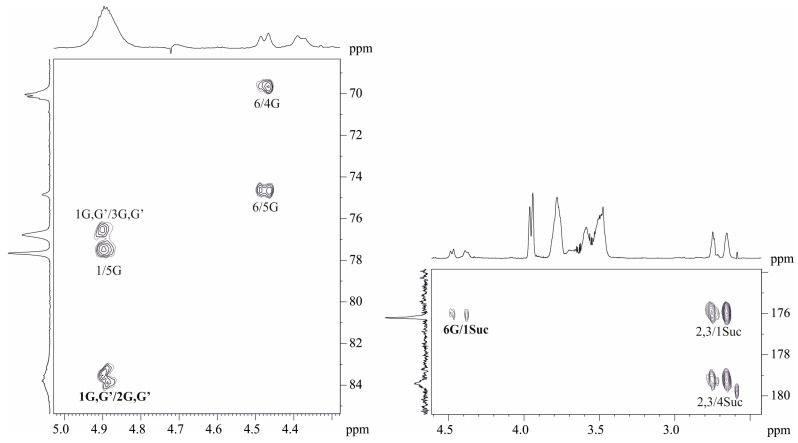
Part of a 2D ^1^H,^13^C HMBC spectrum of the PS-2 of *O. quorumnocens* T1Kr02. The corresponding part of the ^1^H NMR spectrum is displayed along the horizontal F2 axis. Arabic numerals refer to the H/C pairs in the sugar residues denoted by letters as shown in Table 1.

**Figure 6 ijms-25-01970-f006:**
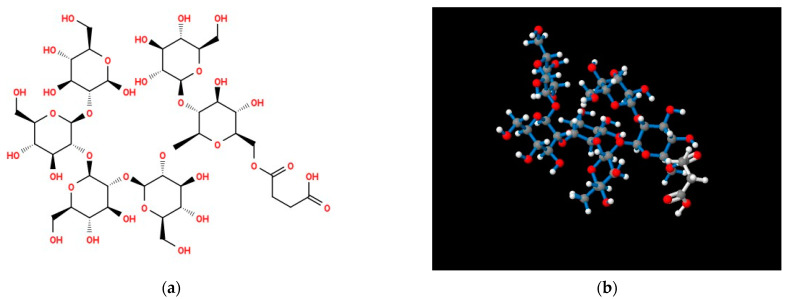
Structure formula (**a**) and molecular model (**b**) of β-(1→2) glucan (PS-2) from *O. quorumnocens* T1Kr02. One of the d-Glc*p* residues (~15%) at position 6 links with a succinate residue. Atoms (balls) are designated as follows: gray—carbon; red—oxygen; white—hydrogen. Bonds between atoms in the oligosaccharide are indicated by blue sticks, and bonds in the succinate group are indicated by white sticks.

**Figure 7 ijms-25-01970-f007:**
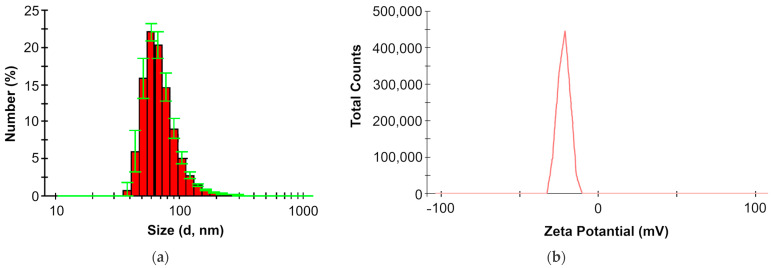
Results of measuring the size (**a**) and zeta-potential (**b**) of supramolecular particles of *O. quorumnocens* T1Kr02 LPS in an aqueous solution (C = 1 mg/mL) at 37 °C by dynamic light scattering.

**Figure 8 ijms-25-01970-f008:**
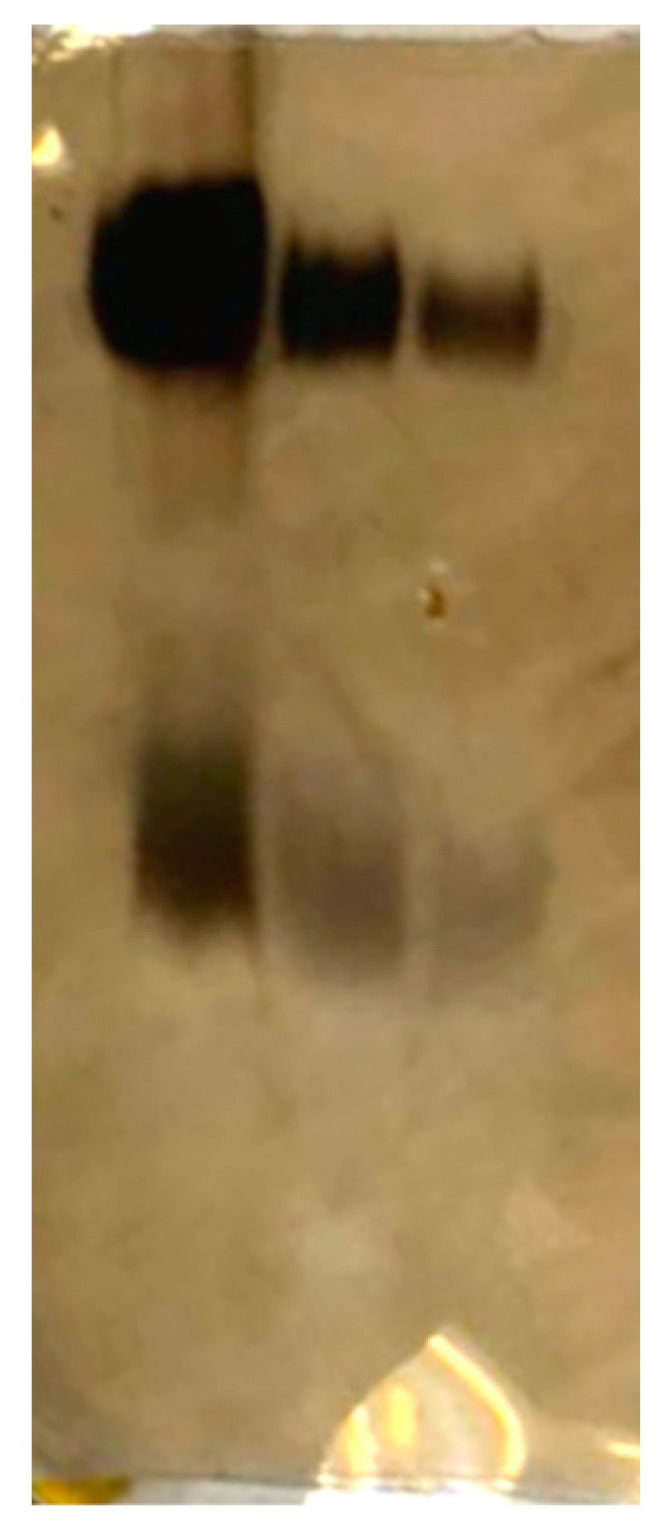
SDS-PAGE result of LPS preparation from *O. quorumnocens* T1Kr02. The amount of LPS per track was 40 μg, 20 μg, and 10 μg.

**Table 1 ijms-25-01970-t001:** ^1^H and ^13^C NMR chemical shifts (δ, ppm) of the polysaccharides of *O. quorumnocens* T1Kr02.

Residue	C-1 *H-1*	C-2 *H-2*	C-3 *H-3*	C-4 *H-4*	C-5 *H-5*	C-6 *H-6*
* **DPS** *						
→2)-β-d-Fuc*f*-(1→ (***A***)	109.0 *5.43*	90.5 *4.34*	77.9 *4.10*	88.4 *3.88*	68.6 *3.94*	19.5 *1.26*
→3)-β-d-Fuc*p*-(1→ (***B***)	103.3 *4.60*	70.8 *3.60*	81.4 *3.74*	72.5 *3.92*	72.1 *3.83*	16.7 *1.26*
* **PS-1** *						
→2)-β-d-Fuc*f*-(1→ (***A***)	108.8 *5.43*	90.5 *4.35*	77.7 *4.11*	88.3 *3.88*	68.6 *3.89*	19.5 *1.26*
→3)-β-d-Fuc*p*-(1→ (***B′***)	102.9 *4.67*	70.9 *3.60*	81.0 *3.77*	72.3 *3.90*	72.2 *3.80*	16.6 *1.26*
→2)-β-d-Fuc*f*-(1→ (***A′***) 3) | OAc	109.0 *5.49* 174.6	87.0 *4.46* 21.6 *2.16*	79.3 *5.04*	87.3 *4.14*	68.3 *4.02*	19.5 *1.23*
→3)-β-d-Fuc*p*-(1→ (***B***)	103.4 *4.60*	70.9 *3.62*	81.4 *3.74*	72.3 *3.90*	72.2 *3.83*	16.6 *1.27*
* **PS-2** *						
→2)-α-d-Glc*p*-(1→ (***G***)	103.5 *4.90*	83.8 *3.59*	76.8 *3.79*	70.1 *3.48*	77.8 *3.51*	62.1 *3.95*, *3.77*
→2)-α-d-Glc*p*-(1→ (***G′***) 6) | 1)-Suc	103.5 *4.90* 176.2	83.8 *3.59* 30.9 * *2.75*	76.9 *3.67* 31.2 * *2.66*	69.9 *3.59* 179.4	74.8 *3.70*	64.5 *4.48*, *4.39*

^1^H NMR chemical shifts are shown in italics. * The assignment can be reversed.

**Table 2 ijms-25-01970-t002:** Correlations for H-1 and C-1 in the 2D ROESY and HMBC spectra of the *Ochrobactrum quorumnocens* T1Kr02 polysaccharides.

Anomeric Atom in Sugar Residue (δ)	Correlation(s) to Atoms in Sugar Residue(s) (δ)	
	ROESY	HMBC
**DPS**		
**A** H-1 (5.43)	**A** H-2 (4.34), **B** H-3 (3.74), **B** H-2 (3.60)	**A** C-3 (77.6), **B** C-3 (814), **A** C-4 (88.4), **A** C-2 (90.5)
**A** C-1 (109.0)		**A** H-2 (4.34), **A** H-3 (4.10), **B** H-3 (3.74)
**B** H-1 (4.60)	**A** H-2 (4.34), **A** H-3 (4.10), **B** H-5 (3.83), **B** H-3 (3.74), **B** H-2 (3.60)	**B** C-2 (90.5)
**B** C-1 (103.3)		**A** H-2 (4.34), **B** H-5 (3.83), **B** H-2 (3.60)
**PS-1**		
**A** H-1 (5.43)	**A** H-2 (4.34), **B′** H-4 (3.90), **B′** H-3 (3.77), **B′** H-2 (3.60)	**A** C-3 (77.7), **B′** C-3 (81.4), **A** C-4 (88.3)
**A** C-1 (108.8)		**A** H-2 (4.34), **B′** H-3 (3.77)
**B′** H-1 (4.67)	**A′** H-2 (4.46), **B′** H-3 (3.77), **B′** H-5 (3.80), **B′** H-2 (3.60)	**A′** C-2 (87.0)
**B′** C-1 (102.9)		**A′** H-2 (4.45), **B′** H-5 (3.80), **B**’ H-2 (3.60)
**A′** H-1 (5.49)	**A′** H-2 (4.46), **B** H-4 (3.90), **B** H-3 (3,74), **B** H-2 (3.60)	**A′** C-3 (79.3), **A′** C-4 (87.3), **B** C-3 (81.4)
**A′** C-1 (109.0)		**A′** H-2 (4.46), **B** H-3 (3.77)
**B** H-1 (4.60)	**A** H-2 (4.34), **A** H-3 (4.11), **B** H-5 (3.83), **B** H-3 (3.74)	**A** C-2 (90.5)
**B** C-1 (103.4)		**A** H-2 (4.35), **B** H-5 (3.83), **B** H-2 (3.62)

**Table 3 ijms-25-01970-t003:** Morphometric variables of 30-day-old potato (cv. Kondor) microplants after incubation with LPS of *Ochrobactrum quorumnocens* T1Kr02.

Variable	Control	Experiment (+LPS)
Shoot length, mm	59.0 ± 3.9 *	**79.8 ± 6.8** **
Node number, pcs	7.85 ± 0.39	6.90 ± 0.55
Root number, pcs	10.0 ± 0.66	10.4 ± 0.53
Sum of root length, cm	240.6 ± 33.1	304.7 ± 33.9
Wet weight of shoots, mg	304.7 ± 23.4	334.3 ± 8.1
Dry weight of shoots, mg	31.5 ± 2.1	**55.3 ± 11.4** **
Wet weight of roots, mg	89.2 ± 15.3	**168.7 ± 23.6** ***
Dry weight of roots, mg	9.76 ± 0.92	**15.33 ± 2.33** **

*—confidence intervals for *p* = 0.05; **—values in bold differ significantly from the control for *p* = 0.05; ***—the values differ significantly from the control for *p* = 0.01.

## Data Availability

Data are contained within the article and its Appendix A. Additional data supporting the findings of this study are available from the corresponding author upon reasonable request.

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
