# Peer review of "Structure, Physicochemical Properties and Biological Activity of Lipopolysaccharide from the Rhizospheric Bacterium Ochrobactrum quorumnocens T1Kr02, Containing d-Fucose Residues"

_ijms, 2024, doi:10.3390/ijms25041970_

Round 1

Reviewer 1 Report

Comments and Suggestions for Authors

The manuscript titled "Structure, Physicochemical Properties and Biological Activity of Lipopolysaccharide from the Rhizospheric Bacterium Ochrobactrum quorumnocens T1Kr02, Containing D-fucose Residues" provides valuable insights into the characterization of lipopolysaccharides (LPS) from the bacterium Ochrobactrum quorumnocens T1Kr02. The study involves the identification of the O-polysaccharide structure, fatty acid analysis, physicochemical properties, and biological activity of the LPS.

The O-polysaccharide structure of O. quorumnocens T1Kr02 was identified using NMR, revealing a unique repeating unit: →2)--D-Fucf-(1→3)--D-Fucp-(1→.Non-stoichiometric modifications were identified, with 50% of D-Fucf residues at position 3 being O-acetylated, and 15% of D-Glcp residues at position 6 being linked with succinate.

This is the first report of a polysaccharide containing both D-fucopyranose and D-fucofuranose residues.

Fatty acid analysis of the LPS revealed the prevalence of specific acids, including 3-hydroxytetradecanoic, hexadecenoic, octadecenoic, lactobacillic, and 27-hydroxyoctacosanoic acids.

Comparative analysis with other strains and genera provides a basis for understanding the unique features of O. quorumnocens T1Kr02 LPS.

The manuscript is well-structured, and the methods used for characterization are appropriate. The detailed analysis of the polysaccharide structure, fatty acid composition, and biological activity contributes significantly to the existing knowledge in this field. The results presented in the manuscript have implications for both fundamental research in microbiology and potential applications in agriculture.

Minor revisions may be considered for further clarification of certain technical details and for improving the flow of the manuscript: 

- As the genome of strain T1Kr02 has not been deciphered, consider discussing the potential genetic basis of the biosynthesis of D-fucose residues and how it compares to other known pathways.

-The manuscript mentions the need for a special biochemical study to clarify enzyme activities. Consider in conclusions part elaborating on the importance of such a study and potential future directions.

-Additionally, the manuscript could benefit from a more explicit discussion of the potential applications of the findings in industry biotechnology.

Overall, this manuscript provides valuable information on the structural and functional aspects of Ochrobactrum quorumnocens T1Kr02 LPS, making it a valuable contribution in the field of microbiology and biotechnology.

Author Response

Reviewer’s comment:

The manuscript titled "Structure, Physicochemical Properties and Biological Activity of Lipopolysaccharide from the Rhizospheric Bacterium Ochrobactrum quorumnocens T1Kr02, Containing D-fucose Residues" provides valuable insights into the characterization of lipopolysaccharides (LPS) from the bacterium Ochrobactrum quorumnocens T1Kr02. The study involves the identification of the O-polysaccharide structure, fatty acid analysis, physicochemical properties, and biological activity of the LPS.

The O-polysaccharide structure of O. quorumnocens T1Kr02 was identified using NMR, revealing a unique repeating unit: →2)-β-D-Fucf-(1→3)-β-D-Fucp-(1→. Non-stoichiometric modifications were identified, with 50% of D-Fucf residues at position 3 being O-acetylated, and 15% of D-Glcp residues at position 6 being linked with succinate.

This is the first report of a polysaccharide containing both D-fucopyranose and D-fucofuranose residues.

Fatty acid analysis of the LPS revealed the prevalence of specific acids, including 3-hydroxytetradecanoic, hexadecenoic, octadecenoic, lactobacillic, and 27-hydroxyoctacosanoic acids.

Comparative analysis with other strains and genera provides a basis for understanding the unique features of O. quorumnocens T1Kr02 LPS.

The manuscript is well-structured, and the methods used for characterization are appropriate. The detailed analysis of the polysaccharide structure, fatty acid composition, and biological activity contributes significantly to the existing knowledge in this field. The results presented in the manuscript have implications for both fundamental research in microbiology and potential applications in agriculture.

Minor revisions may be considered for further clarification of certain technical details and for improving the flow of the manuscript: 

Response:

We are thankful to the reviewer for closely reading our manuscript and leaving comments. Our responses are summarized below. All corrections made to the manuscript are color-coded.

Reviewer’s comment:

- As the genome of strain T1Kr02 has not been deciphered, consider discussing the potential genetic basis of the biosynthesis of D-fucose residues and how it compares to other known pathways.

Response:

Thank you for this comment. We have added a description of the potential genetics of fucose residue biosynthesis in strain T1Kr02. In the revised manuscript, this fragment is:

In E. coli O52, dTDP-D-Fucf was shown to be synthesized from dTDP-d-Fucp by the Fcf2 mutase [49] (Figure S4). Since the genome of strain T1Kr02 has not yet been deciphered, we do not know the genetics of the O-polysaccharide biosynthesis of this strain. However, based on the monosaccharide composition, we predict that the O-antigen biosynthesis gene cluster of the strain T1Kr02 contains genes for the biosynthesis of nucleotide-activated d-fucopyranose (rmlA, rmlB, and fcf1) and d-fucofuranose (rmlA, rmlB, fcf1, and fcf2).

  1. Wang, Q.; Ding, P.; Perepelov, A.V.; Xu, Y.; Wang, Y.; Knirel, Y.A.; Wang, L.; Feng, L. Characterization of the dTDP‐d‐fucofuranose biosynthetic pathway in Escherichia coli O52. Mol. Microbiol. 2008, 70(6), 1358-1367. https://doi.org/10.1111/j.1365-2958.2008.06449.x

We have also changed the manuscript text to show more clearly than the enzyme competition problem we described for dTDP-d-Fucp in the synthesis of the O-polysaccharide of strain T1Kr02 is a hypothesis and is only possible if dTDP-d-Fucp is also a precursor of dTDP-d-Fucf in strain T1Kr02.

To our knowledge, only one single pathway for the biosynthesis of nucleotide-activated d-fucose residues has been described in bacteria to date (IUBMB; Wang et al., 2008): from glucose-1-phosphate, via dTDP-α-d-Glcp and dTDP-4-dehydro-6-deoxy-α-d-Glcp to dTDP-d-Fucp. This scheme is presented in the manuscript in Figure S4. For plants, the biosynthesis pathway of nucleotide-activated d-fucose residues is unknown. However, it is assumed that the same reactions take place in plants as in bacteria, but instead of dTDP, plant enzymes use UDP (Reed et al., 2023).

IUBMB - International Union of Biochemistry and Molecular Biology, https://iubmb.qmul.ac.uk/enzyme/reaction/polysacc/dTDPdeoxyhex.html

Wang, Q.; Ding, P.; Perepelov, A.V.; Xu, Y.; Wang, Y.; Knirel, Y.A.; Wang, L.; Feng, L. Characterization of the dTDP‐d‐fucofuranose biosynthetic pathway in Escherichia coli O52. Mol. Microbiol. 2008, 70(6), 1358-1367. https://doi.org/10.1111/j.1365-2958.2008.06449.x

Reed, J.; Orme, A.; El-Demerdash, A.; Owen, C.; Martin, L.B.; Misra, R.C.; Kikuchi, S.; Rejzek, M.; Martin, A.C.; Harkess, A.; Leebens-Mack, J. Elucidation of the pathway for biosynthesis of saponin adjuvants from the soapbark tree. Science. 2023, 379(6638), 1252-1264. https://doi.org/10.1126/science.adf3727

Reviewer’s comment:

-The manuscript mentions the need for a special biochemical study to clarify enzyme activities. Consider in conclusions part elaborating on the importance of such a study and potential future directions.

Response:

Thank you for this remark. We have included in the Conclusion section a discussion of the importance of biochemical studies of enzymatic activity for the development of biotechnology for the synthesis of natural glycans.

Reviewer’s comment:

-Additionally, the manuscript could benefit from a more explicit discussion of the potential applications of the findings in industry biotechnology.

Response:

We appreciate your opinion and have added the following text to the Discussion section:

LPS and O-polysaccharide of strain T1Kr02, due to their unique chemical composition and structure, can potentially also find applications in industrial biotechnology. Polymers containing 6-deoxyhexose residues can be used due to their rheological properties such as gelation, viscosity, or bioemulsification, and they also represent a source of some important monosaccharides [66]. The industrially important properties of d-Fuc-containing polysaccharides are still poorly understood, but these glycans can become a profitable source of various forms of d-Fuc and their modified derivatives.

  1. Paul, F.; Morin, A.; Monsan, P. Microbial polysaccharides with actual potential industrial applications. Biotechnol. Adv. 1986, 4(2), 245-259. https://doi.org/10.1016/0734-9750(86)90311-3

Reviewer’s comment:

Overall, this manuscript provides valuable information on the structural and functional aspects of Ochrobactrum quorumnocens T1Kr02 LPS, making it a valuable contribution in the field of microbiology and biotechnology.

Response:

We appreciate the reviewer commending our manuscript and hope that the manuscript will be better after the revision is done.

Reviewer 2 Report

Comments and Suggestions for Authors

This is a good work and can be published with minor revisions.

The authors note (lines 354-355) that one of the mechanisms of the effect of LPS on plants is an increase in peroxidase activity and an effect on calcium channels. However, how can LPS interact with plants? Do plants have LPS receptors?

The Conclusions section should be revised as it only contains perspectives. Typically, conclusions also include the main results obtained.

Small comments:

I think the plural form LPS (LPSs) could be omitted. The authors used LPSs only twice (lines 54 and 224), compared to 68 times for LPS. It is not clear why LPSs is used rather than LPS in some cases.

Author Response

Reviewer’s comment:

This is a good work and can be published with minor revisions.

Response:

We appreciate the reviewer commending our manuscript and hope that the manuscript will be better after the revision is done. Our responses are summarized below. All corrections made to the manuscript are color-coded.

Reviewer’s comment:

The authors note (lines 354-355) that one of the mechanisms of the effect of LPS on plants is an increase in peroxidase activity and an effect on calcium channels. However, how can LPS interact with plants? Do plants have LPS receptors?

Response:

Thank you for the test incorrectness you pointed out. We have changed that part of the manuscript. Instead of

“One of the mechanisms of influence on plants is an increase in peroxidase activity and a change in the functioning of calcium channels.”

we’ve added the next text:

“The molecular mechanisms of interaction between bacterial LPS and plant cells are still poorly studied. To date, only two plant receptors have been described to participate in the formation of responses to LPS or lipid A: in Arabidopsis thaliana, the bulb-type lectin receptor-like kinase AtLORE [61], and in Oryza sativa, the LysM-type receptor-like kinase OsCERK1 [62]. At the same time, it is assumed that plants also have other receptors for LPS [21]. One of the mechanisms of specific plant responses to LPS from phytopathogenic or growth-promoting bacteria may be differences in changes in peroxidase activity and the functioning of calcium channels [63].”

21. Kutschera, A.; Ranf, S. The multifaceted functions of lipopolysaccharide in plant-bacteria interactions. Biochim. 2019, 59, 93-98. https://doi.org/10.1016/j.biochi.2018.07.028

61. Ranf, S.; Gisch, N.; Schäffer, M.; Illig, T.; Westphal, L.; Knirel, Y.A.; Sánchez-Carballo, P.M.; Zähringer, U.; Hückelhoven, R.; Lee. J.; Scheel, D. A lectin S-domain receptor kinase mediates lipopolysaccharide sensing in Arabidopsis thaliana. Nat. Immunol. 2015, 16(4), 426-433. https://doi.org/10.1038/ni.3124

62. Desaki, Y.; Kouzai, Y.; Ninomiya, Y.; Iwase, R.; Shimizu, Y.; Seko, K.; Molinaro, A.; Minami, E.; Shibuya, N.; Kaku, H.; Nishizawa, Y. OsCERK1 plays a crucial role in the lipopolysaccharide‐induced immune response of rice. New Phytol. 2018, 217(3), 1042-1049. https://doi.org/10.1111/nph.14941

63. Hernández-Esquivel, A.A.; Castro-Mercado, E.; García-Pineda, E. Comparative effects of Azospirillum brasilense Sp245 and Pseudomonas aeruginosa PAO1 lipopolysaccharides on wheat seedling growth and peroxidase activity. J. Plant Growth Regul. 2021, 40, 1903-1911. https://doi.org/10.1007/s00344-020-10241-x

Reviewer’s comment:

The Conclusions section should be revised as it only contains perspectives. Typically, conclusions also include the main results obtained.

Response:

We appreciate your feedback. In the Conclusion section we’ve added the main results obtained.

Reviewer’s comment:

Small comments: I think the plural form LPS (LPSs) could be omitted. The authors used LPSs only twice (lines 54 and 224), compared to 68 times for LPS. It is not clear why LPSs is used rather than LPS in some cases.

Response:

Thanks for noticing this. This is our mistake. In the revised manuscript, “LPS” is written in all cases.